# Rhamnopyranoside-Based Fatty Acid Esters as Antimicrobials: Synthesis, Spectral Characterization, PASS, Antimicrobial, and Molecular Docking Studies

**DOI:** 10.3390/molecules28030986

**Published:** 2023-01-18

**Authors:** Abul Fazal Muhammad Sanaullah, Puja Devi, Takbir Hossain, Sulaiman Bin Sultan, Mohammad Mohib Ullah Badhon, Md. Emdad Hossain, Jamal Uddin, Md. Abdul Majed Patwary, Mohsin Kazi, Mohammed Mahbubul Matin

**Affiliations:** 1Bioorganic and Medicinal Chemistry Laboratory, Department of Chemistry, Faculty of Science, University of Chittagong, Chittagong 4331, Bangladesh; 2Wazed Miah Science Research Centre (WMSRC), Jahangirnagar University, Savar, Dhaka 1342, Bangladesh; 3Center for Nanotechnology, Department of Natural Sciences, Coppin State University, Baltimore, MD 21216, USA; 4Department of Chemistry, Comilla University, Cumilla 3506, Bangladesh; 5Department of Pharmaceutics, College of Pharmacy, King Saud University, P.O. Box 2457, Riyadh 11451, Saudi Arabia

**Keywords:** di-*O*-stearate, in vitro test, methyl α-l-rhamnopyranoside, molecular docking, regioselective acylation, structure-activity relationship, sugar esters

## Abstract

The most widely used and accessible monosaccharides have a number of stereogenic centers that have been hydroxylated and are challenging to chemically separate. As a result, the task of regioselective derivatization of such structures is particularly difficult. Considering this fact and to get novel rhamnopyranoside-based esters, DMAP-catalyzed di-*O*-stearoylation of methyl α-l-rhamnopyranoside (**3**) produced a mixture of 2,3-di-*O*- (**4**) and 3,4-di-*O*-stearates (**5**) (ratio 2:3) indicating the reactivity of the hydroxylated stereogenic centers of rhamnopyranoside as 3-OH > 4-OH > 2-OH. To get novel biologically active rhamnose esters, di-*O*-stearates **4** and **5** were converted into six 4-*O*- and 2-*O*-esters **6**–**11**, which were fully characterized by FT-IR, ^1^H, and ^13^C NMR spectral techniques. In vitro antimicrobial assays revealed that fully esterified rhamnopyranosides **6**–**11** with maximum lipophilic character showed better antifungal susceptibility than antibacterial activity. These experimental findings are similar to the results found from PASS analysis data. Furthermore, the pentanoyl derivative of 2,3-di-*O*-stearate (compound **6**) showed better antifungal functionality against *F. equiseti* and *A. flavus*, which were found to be better than standard antibiotics. To validate the better antifungal results, molecular docking of the rhamnose esters **4**–**11** was performed with lanosterol 14α-demethylase (PDB ID: 3LD6), including the standard antifungal antibiotics ketoconazole and fluconazole. In this instance, the binding affinities of **10** (−7.6 kcal/mol), **9** (−7.5 kcal/mol), and **7** (−6.9 kcal/mol) were better and comparable to fluconazole (−7.3 kcal/mol), indicating the likelihood of their use as non-azole type antifungal drugs in the future.

## 1. Introduction

Due to their extremely specialized interactions with physiological receptors, carbohydrate molecules take part in a wide range of biological activities [1]. Their esters, particularly monosaccharide-based sugar esters (SEs), play a variety of roles in biological processes involving species from all walks of life [1,2]. SEs are made up of a carbohydrate moiety (hydrophilic) and one or more fatty acid portions (lipophilic moieties), which probably contribute to their non-toxic, biodegradable, non-allergic, non-irritating, fat replacer, and emulsifier properties [3,4]. SEs have generated a great deal of scientific attention and have a wide range of applications in both industry and medicine due primarily to their significant insecticidal and antibacterial capabilities [5,6,7,8]. Notably, SEs’ hydrophobic chains can influence cytotoxicity against human skin melanoma and prostate cancer cell lines [5]. In the medical and pharmaceutical fields, SEs and associated medications have also opened a new door for drug delivery. These drugs often depend on (i) sugar linkage, (ii) esterified/protected hydroxyl group(s), and (iii) unesterified/unprotected hydroxyl group(s) [9,10]. These elements are known to increase a substance’s aqueous solubility, stability, and biocompatibility, which helps the medicine reach its intended target site and improves absorption [11]. For instance, compared to aspirin alone, the anticancer activity and water solubility of the glucose-aspirin ester were improved eight to nine times and seven times, respectively [12]. Positively, some of the SEs were reported to be extremely effective against pathogenic organisms that are multidrug-resistant (MDR) [13,14].

Among the monosaccharides, L-rhamnopyranose is a widely distributed natural carbohydrate [15,16]. For instance, brasilicardin A (**1**, Figure 1), an actinomycete with two rhamnopyranosyl moieties, was present in the broth of the actinomycete *Nocardia brasiliensis* IFM0406 [17]. The incorporation of the hydroxybenzoyl group in the rhamnose unit of **1** enhances its immunosuppressive properties. Moreover, rhamnose-based ester **2** isolated from *S. buergeriana* exhibited glutamate-induced neurotoxicity [18]. All of these findings suggested that the esterification of the rhamnose moiety in natural products is crucial for the creation of potent anticancer medicines (RSK inhibitors) with high-affinity binding and selectivity [19]. Thus, rhamnopyranoside-based ester compounds got special focus in research related to antigenic, anticarcinogenic [19,20], antimicrobial [21,22,23,24,25,26,27,28], and pharmacological properties [29].

Despite a plethora of opportunities for rhamnose esters, it seems challenging for site-selective esterification/acylation of its three secondary OH groups of similar reactivity [21,30,31]. Various methods for obtaining selectively acylated rhamnopyranosides have been reported [32]. Previously, our group reported selective 3-*O*- [26] and 4-*O*-monoacylation [23,27] of rhamnopyranoside. Thus, we were interested in investigating the di-*O*-acylation of rhamnopyranoside **3** using 4-dimethylaminopyridine (DMAP) as a catalyst. Although sugar fatty acid esters have been known for a long time and extensive study has been done on their application features [33], many concerns about the nature of these molecules remain unanswered [34], leaving room for further scientific investigation and practical studies. In this respect, in vitro antimicrobial tests, ADMET, and molecular docking of the synthesized rhamnopyranoside esters were conducted and reported herein.

## 2. Results and Discussion

### 2.1. Selective Stearoylation: Synthesis of Methyl 2,3-di-O-stearoyl-α-l-rhamnopyranoside (***4***) and Methyl 3,4-di-O-stearoyl-α-l-rhamnopyranoside (***5***)

Encouraged by a number of interesting and effective results we have chosen methyl α-l-rhamnopyranoside (**3**), a representative compound for regioselective acylation using stearoyl chloride. The synthesized novel rhamnopyranoside stearoyl esters were further subjected to acylation reactions at the free OH positions (secondary hydroxyl groups) with various acylating agents for the purpose of synthesizing newer compounds for structure elucidation and the search for new antimicrobial agents.

At first, a DMAP catalyst was used to react methyl α-l-rhamnopyranoside (**3**) with dimolar stearoyl chloride in pyridine at 0 °C. The mixture was allowed to attain room temperature after 8 h and stirred for an additional 10 h at room temperature when TLC indicated the formation of two faster-moving products (Figure 1). Usual work-up and chromatography gave two semi-solids with higher *R*_f_ (0.68) compound **4** (26%) as a syrup and lower *R*_f_ (0.33) compound **5** (39%) as a semi-solid mass (ratio of the mixture 2:3), which resisted crystallization.

The semi-solid, with a higher *R*_f_ (0.68), resonated at 3300–3500 (br, OH), 1744, 1739 (CO), and 1065 cm^−1^ (pyranose ring) in its FT-IR spectrum (Appendix A), and hence indicated the attachment of two stearoyl groups to the rhamnopyranoside molecule. In its ^1^H NMR spectrum (Appendix A), two two-proton triplets at δ 2.38 and 2.31 (*J* = 7.6 Hz), one four-proton multiplet at δ 1.59–1.66, one fifty-six-proton broad multiplet at δ 1.20–1.37, and a six-proton triplet at δ 0.89 (*J* = 6.4 Hz) totaling seventy additional protons (70H) indicated the attachment of two stearoyl groups in the molecule. More informatively, H-2 at δ 5.23 (s) and H-3 at δ 5.12–5.15 (dd, *J* = 9.6 and 3.2 Hz) were found to resonate very downfield as compared to its precursor, rhamnopyranoside **3** [21]. This clearly demonstrated the incorporation of the stearoyloxy group at the C-2 and C-3 positions. This fact was further confirmed by analyzing its ^13^C NMR spectrum (Appendix A), which exhibited two carbonyl carbons at δ 174 and 172 positions and thirty-four aliphatic carbon signals in addition to methyl α-l-rhamnopyranoside carbons. The structure and position of the signals were also confirmed by its DEPT-135, 2D COSY, and 2D HMBC spectra (Figure 2A and Appendix A). Compiling all these spectral data confirmed the structure of the higher *R*_f_ compound as methyl 2,3-di-*O*-stearoyl-α-l-rhamnopyranoside (**4**).

Further elution of the reaction mixture with chloroform-methanol (15:1) slowly furnished the lower *R*_f_ compound as a semi-solid. In its ^1^H NMR spectrum (Appendix A), a four-proton multiplet at δ 2.23–2.36, one four-proton multiplet at δ 1.58–1.70, one broad fifty-six-proton multiplet at δ 1.25–1.38, and a six-proton triplet at δ 0.90 (*J* = 6.4 Hz) indicated the attachment of two stearoyl groups (additional 70H) in the molecule. Also, H-3 proton (at 5.23–5.26, dd, *J* = 9.5 and 3.2 Hz) and H-4 (at 5.13, t, *J* = 10.0 Hz) resonated considerably downfield as compared to its precursor methyl α-l-rhamnopyranoside (**3**) [21]. This clearly confirmed the incorporation of the stearoyloxy group at the C-3 and C-4 positions. This fact was further confirmed by analyzing its ^13^C NMR spectrum (Appendix A), which exhibited characteristic two carbonyl-carbons at δ 174.0, and 172.8 and thirty-four aliphatic carbon signals in addition to usual methyl α-l-rhamnopyranoside carbons. The structure and position of the signals were also confirmed by the 2D HMBC spectrum (Figure 2B and Appendix A). As a result, the compound with the lower *R*_f_ was established as methyl 3,4-di-*O*-stearoyl-α-l-rhamnopyranoside (**5**).

Thus, the DMAP catalyzed dimolar stearoylation (2.2 eq. amount) of methyl α-l-rhamnopyranoside at low temperatures (0–25 °C) in pyridine showed regioselectivity at C-2, C-3, and C-4 positions, indicating that the reactivities of OH groups in methyl α-l-rhamnopyranoside (**3**) are 3-OH ˃ 4-OH ˃ 2-OH.

### 2.2. 4-O-Acylation of Di-O-stearate ***4***

The free OH group present in the di-*O*-stearate **4** is exploited for further acylation with different acylating agents. Initially, hexanoyl chloride was added to a solution of di-*O*-stearate **4** in dry pyridine, and then, as usual, the work-up procedure provided clear syrup in 79% (Figure 2).

The FT-IR spectrum (Appendix A) of this syrup showed four CO bands at 1748, 1747, and 1726 cm^−1^ and the absence of OH bands. Thus, it indicates the attachment of one hexanoyl and two stearoyl groups to the molecule. In the ^1^H NMR spectrum (Appendix A), H-4 resonated at δ 5.1 (t, *J* = 10.0 Hz) (Table 1) and this value appeared at more downfield region than its precursor compound **4** (δ 3.60). The other ^1^H NMR signals were found as three six-proton multiplets at δ 2.19–2.43, a twelve-proton multiplet at δ 1.55–1.67, a fifty-seven-proton broad multiplet at δ 1.23–1.32 (with the three-methyl proton at the C-6 position), and a nine-proton multiplet at δ 0.88–0.91. These data indicated that one hexanoyl group was attached to this compound. Also, in its ^13^C NMR spectrum (Appendix A), it showed three signals at δ 172.9, 172.7, and 172.6 positions, indicating the attachment of three carbonyl groups in this compound. Furthermore, this compound exhibited thirty-eight aliphatic carbon signals in the ^13^C NMR spectrum in addition to regular rhamnopyranoside carbons. A complete analysis of its FT-IR, ^1^H, and ^13^C NMR established the structure as methyl 4-*O*-hexanoyl-2,3-di-*O*-stearoyl-α-l-rhamnopyranoside (**6**).

In the next step, di-*O*-stearate compound **4** was converted into methyl 4-*O*-decanoyl-2,3-di-*O*-stearoyl-α-l-rhamnopyranoside **(7)** by employing decanoyl chloride. The FT-IR spectrum (Appendix A) of this compound showed no stretching peaks for the free OH group. This clearly indicates the full esterification of this compound. In its ^1^H NMR spectrum (Appendix A), H-4 resonated at δ 5.10 (t, *J* = 9.6 Hz) (Table 1), and this downfield value of H-4 compared to its precursor compound **4** (δ 3.60) indicated the attachment of decanoyl group at the C-4 position. The other ^1^H NMR signals appeared as a six-proton multiplet at δ 2.39–2.46, a four-proton multiplet at δ 2.19–2.30, a twelve-proton multiplet at δ 1.55–1.65, a sixty-one-proton broad multiplet at δ 1.23–1.32 (including three methyl-protons at the C-6 position), and a nine-proton triplet at δ 0.88–0.91 corresponding to the formation of methyl 4-*O*-decanoyl-2,3-di-*O*-stearoyl-α-l-rhamnopyranoside (**7**). This structure of compound **7** was further confirmed with its ^13^C NMR spectrum (Appendix A).

Finally, di-*O*-stearate **4** was treated with benzoyl chloride (1.1 eq.) in DMAP at 0 °C and stirred at room temperature for 14 h, which furnished a syrupy product (Figure 2). In its ^1^H NMR spectrum (Appendix A), three multiplets at δ 7.86–8.06, 7.48–7.59, and 7.32–7.46 indicate the resonance for five aromatic protons. Furthermore, H-4 resonated at δ 5.35 (t, *J* = 6.4 Hz) (Table 1), and the downfield value of H-4 compared to its precursor compound **4** (δ 3.60) indicated the attachment of the benzoyl group at the C-4 position. Again, in the ^13^C NMR spectrum (Appendix A), three carbonyl carbon signals appeared at δ 172.9, 172.6, and 165.6. This demonstrated that three acyl groups were attached to this compound. Furthermore, other carbon signals for one benzoyl group and aliphatic carbon signals in the ^13^C NMR spectrum, in addition to methyl α-l-rhamnopyranoside carbons in their respective positions, clearly indicate the formation of methyl 4-*O*-benzoyl-2,3-di-*O*-stearoyl-α-l-rhamnopyranoside (**8**). This structure was also confirmed by its 2D-COSY spectrum (Appendix A).

### 2.3. 2-O-Acylation of Di-O-stearate ***5***

Having methyl 3,4-di-*O*-stearoyl-α-l-rhamnopyranoside (**5**) in hand, we attempted its derivatization using four different types of acylating agents, such as hexanoyl chloride, decanoyl chloride, and benzoyl chloride, using the direct acylation technique. In compound **5,** the C-2 hydroxyl group is free. We have exploited the position for further esterification which confirmed the synthesized structure 3,4-di-*O*-stearoyl-α-l-rhamnopyranoside (**5**) and provided some novel biologically potential rhamnopyranoside esters.

Firstly, 3,4-di-*O*-stearate **5** on reaction with hexanoyl chloride (1.1 eq.) in anhydrous pyridine in the presence of DMAP (cat.) for 18 h afforded a faster-moving single product as a clear syrup with an 81% yield (Figure 3).

Its FT-IR spectrum (Appendix A) had no OH stretching bands, indicating hexanoylation of the molecule. In the ^1^H NMR spectrum (Appendix A) of this syrup, a three-proton singlet at δ 3.40 was assigned for anomeric OC*H*_3_ protons. The H-2 proton signal appeared at δ 5.26 (as a singlet) as compared to δ 4.04 of its precursor, 3,4-di-*O*-stearate **5** (Table 1). This downfield shift of the H-2 proton was indicative of the attachment of a hexanoyl group at the C-2 position. Again, in the ^13^C NMR spectrum (Appendix A), three carbonyl carbon peaks appeared at δ 172.93, 172.72, and 172.61, demonstrating that three acyl groups are attached to this compound. Furthermore, this compound exhibited forty aliphatic carbon signals in the ^13^C NMR spectrum in addition to the usual rhamnopyranoside carbons. Based on the above ^1^H and ^13^C NMR data, the structure of the compound was assigned as methyl 2-*O*-hexanoyl-3,4-di-*O*-stearoyl-α-l-rhamnopyranoside (**9**).

Similarly, decanoylation and benzoylation of distearate **5** furnished the corresponding 2-*O*-acylates **10** (Appendix A) and **11** (Appendix A), respectively, in good yield (Figure 3). As shown in Table 1, in compounds **10**–**11**, the H-2 proton shifted to a considerably higher frequency as compared to its precursor compound **5**.

### 2.4. Prediction of Activity Spectra for Substances (PASS) of ***3**–**11***

Web-based PASS (prediction of activity spectra for substances; http://www.pharmaexpert.ru/PASSonline/index.php; accessed on 1 November 2022) [35,36] was used for the prediction of the biological potential of the compounds. The results are listed as Pa (probability for an active compound) and Pi (probability for an inactive compound). In the present study, Pa and Pi for different stearates (**4**–**11**) are summarized in Table 2.

PASS biological analysis (Table 2) indicates 0.57 < Pa < 0.62 for antibacterial and 0.68 < Pa < 0.73 for antifungal, suggesting that the rhamnose stearates **4**–**11** should be more active against fungal organisms than bacterial pathogens. The anti-carcinogenic probability of these rhamnose esters is found to be better (0.53 < Pa > 0.70) than that of standard nystatin (Pa = 0.42). Although the addition of two stearoyl groups increases the anti-carcinogenic properties (as in **5**, Pa = 0.70) in the rhamnopyranoside **3** (Pa = 0.662) skeleton, further incorporation of acyl group(s) decreases anti-carcinogenic potentiality (**6**–**11**, Pa = 0.53–0.57). Finally, the anti-viral (Herpes) probability of these rhamnose esters is low (0.46 < Pa > 0.55) compared to that of standard nystatin (Pa = 0.96). Here, the addition of stearoyl groups and/or further incorporation of acyl group(s) did not increase the anti-viral properties of the synthesized rhamnopyranosides. Based on these results and in vitro tests, the synthesized compounds have been docked for antifungal potential (Section 2.7).

### 2.5. In Vitro Antimicrobial Activities of Rhamnopyranoside Esters ***4**–**11***

**Effects of stearoyl rhamnopyranoside derivatives against bacteria**. In the present study, four human pathogenic bacteria were used as test organisms to detect the antibacterial activities [37] of different rhamnopyranoside esters, as shown in Table 3. Among these human pathogens, two were Gram-positive and two were Gram-negative.

From Table 3, it was found that most of the synthetic compounds were not very active against Gram-positive and Gram-negative bacteria, but some compounds showed good to moderate antibacterial activity. Acylated compound **10** (decanoyl derivative of 3,4-di-*O*-stearate) showed the highest antibacterial activity (*25.00 ± 1.73 mm) against *Staphylococcus aureus*, and this value is comparable to the standard antibiotic (*27.66 ± 0.58 mm). Other synthetic compounds show moderate to weak antibacterial functionality against this pathogen. In the case of *Bacillus cereus*, compounds **7** and **10** showed good antibacterial activity compared to that of tetracycline, but the zone of inhibition values of these compounds are a little bit lower than the standard value (Table 3, Figure 3). All synthetic compounds showed weak antibacterial activity against *Escherichia coli*. Similarly, almost all of the compounds were inactive against *Salmonella typhi*. These in vitro results are in conformity with the PASS calculated results (Table 2).

**Effects against fungi**. Azole drugs like fluconazole and itraconazole are found to be comparatively safer and more effective against several acute fungal infections. However, because of the emergence of drug-resistant fungi, anti-fungal drugs are becoming less effective in the treatment of acute fungal infections [37]. To overcome such a threat, alternative antifungals with enhanced efficacy and fungal-specific adjuvants are essential [38,39]. Thus, the in vitro antifungal activities [40] of the synthesized rhamnopyranoside derivatives were investigated against four pathogenic fungi and listed in Table 4. These are *Aspergillus flavus*, *Aspergillus niger*, *Fusarium equiseti*, and *Penicillium* sp.

The results of the percentage inhibitions of mycelial growth (Table 4) showed that all the stearates **4**–**11** have moderate to good antifungal potential. Most of the synthetic compounds exhibit good antifungal activity against *Penicillium* sp., and the observed percent zone of inhibition was comparable to fluconazole. The synthetic compound 4-*O*-hexanoyl-2,3-di-*O*-stearate **6** showed good inhibition against *Aspergillus flavus*, but other compounds were not much active compared to the standard antifungal drug. Rhamnopyranoside stearates **7**, **10**, and **11** are highly active against *Aspergillus niger* (Figure 4), and among them, compound **10** (2-*O*-decanoyl-3,4-di-*O*-stearate) showed the highest activity (*72.62 ± 1.83%) against this fungal pathogen. In the case of *Fusarium equiseti,* all synthetic compounds showed moderate inhibitory activity against this pathogen. Overall, the antifungal susceptibility of the stearates is better than the antibacterial properties. To check and validate this observation, molecular docking of these compounds was conducted and discussed in Section 2.7.

### 2.6. ADME/T Analysis

Drug research and discovery depend heavily on the study of ADME/Tox (Absorption, Distribution, Metabolism, Elimination, and Toxicity). In the current study, toxicity (does this drug have any toxic effects on body systems or organs) and absorption (how much and how quickly the drug is absorbed), distribution (how the drug is distributed within the body), metabolism (how quickly the drug metabolizes), and elimination (or excretion) (how quickly the drug leaves the body) are predicted and presented in Table 5 [41]. Additionally, Table 5 lists a number of physicochemical and drug-like characteristics.

Considering the physicochemical behavior, all compounds have TPSA values that fall below the acceptable range (<140 Å^2^). They have good GI absorption (84–100%). Esters **4**–**11** are non-substrates of CYP2D6 substrates. However, they are substrates of CYP3A4. Their (**4**–**11**) excretion values are higher than the non-ester rhamnopyranoside **3**.

The entire scientific community is interested in a quick and accurate method for evaluating new compounds’ toxicity. Since in vitro and in vivo approaches are frequently constrained by ethics, time, money, and other resources, in silico toxicity prediction methods play a significant role in the selection of lead drugs and in ADMET research [42]. Consequently, in this investigation, machine learning (ML) models [43] were used to assess the possible hepatotoxicity, carcinogenicity, immunotoxicity, mutagenicity, and cytotoxicity of the stearate esters in order to see how these five toxicological elements of drug discovery and development might be affected. The results (Table 5) highlight that the esters are safer (shown in green and paste color) with respect to drug-induced hepatotoxicity (a significant cause of acute liver failure), carcinogenicity, mutagenicity (abnormal genetic mutations), and cytotoxicity (undesired and desired cell damage). The only drawback is that the synthesized novel stearates have an adverse effect of xenobiotics on the immune system, i.e., immunotoxicity (indicated in red in Table 5). However, the overall toxicity class is 6 (non-toxic; LD_50_ > 5000) and 5 (for compounds **8** and **11**; potentially harmful if swallowed; 2000 LD_50_ 5000). Thus, further studies and research are essential to establishing them as safer drugs.

The drug-likeness [44] of the novel stearates, as shown in Table 5, indicated that they have some violation of the Lipinski rule, Veber’s rule, or Ghosh’s rule, rationally due to the presence of two long stearoyl chains in these molecules.

### 2.7. Molecular Docking Results

In recent years, molecular docking has appeared as a significant tool for drug discovery compared with traditional experimental high-throughput screening. For this, the behavior/interactions of the compounds with target proteins can be studied, providing comprehensive insight into the biochemical behavior in the 3D model [45,46]. In the present study, as the rhamnopyranoside esters showed better antifungal susceptibility compared to antibacterial actions, the synthesized compounds were docked against the 3LD6 protein model. In most cases, antifungal drugs are designed to inhibit lanosterol 14α-demethylase (CYP51, 3LD6) as it plays a crucial step in the conversion of lanosterol to ergosterol in the fungal cell membrane [47]. So, the lanosterol 14α-demethylase related protein 3LD6 was selected for molecular docking of the synthesized compounds, and the results are mentioned in Table 6.

Table 6 shows that the binding affinity of compounds **7** (−6.9 kcal/mol), **9** (−7.5 kcal/mol), and **10** (−7.6 kcal/mol) is higher than that of the other rhamnopyranosides, and these higher binding affinity values are comparable to the standard drug fluconazole (−7.3 kcal/mol). The decanoyl derivative of 2,3-di-*O*-stearate **10** showed the highest binding affinity (−7.6 kcal/mol) among all the synthesized products, and again, this value is also high compared to that of fluconazole (−7.3 kcal/mol).

A careful observation indicated that the attachment of the lipophilic stearoyl group at the C-2 and C-3 positions and decanoyl at the C-4 position of **3** (as in **7**) increases its binding affinity (−6.9 kcal/mol). This is probably due to different non-bonding interactions between proteins and compound **7**. However, the addition of stearoyl groups at the C-3 and C-4 positions and hexanoyl (**9**) or decanoyl (**10**) at the C-4 position increases its binding affinity to an excellent level (−7.5 and −7.6 kcal/mol).

Furthermore, the binding affinity of the 3,4-di-*O*-stearate (**19**) increases gradually (from −4.6 to −7.3 kcal/mol) with the introduction of an acyl group from carbon chain length C5 to C10 at position C-2.

On the other hand, the addition of one benzoyl group at the C-4 position (as in **8**) or at the C-2 position (as in **11**) didn’t improve binding affinity satisfactorily (Table 6). Thus, the best active compounds are **10**, **9**, and **7**. These three compounds showed various bonding and nonbonding interactions with the 3LD6 protein (Figure 5). Non-bond interaction analysis clearly showed that the best-scored compound **10** and standard ketoconazole have pi-alkyl interactions with TYR145. Similarly, compound **9** and ketoconazole have the same pi-alkyl interaction with TYR131 and TYR239. These interactions validated the in vitro antifungal test results and the probable future application of these esters as prospective antifungals.

## 3. Materials and Methods

### 3.1. Materials and Instrumentation

Except as otherwise noted, all of the reagents were purchased from Aldrich and used exactly as received. Using conventional techniques, solvents were filtered or used directly from the store. On Kieselgel GF_254_ plates, thin-layer chromatography (TLC) was carried out, and the plates were heated at 150–200 °C by misting them with 1% methanolic sulphuric acid until color occurred. Under reduced pressure, evaporations were carried out in a Buchi rotary evaporator (R-100, Flawil, Switzerland) at temperatures below 40 °C. Using silica gel G_60_, column chromatography (CC) was performed. The solvent systems used for the TLC and CC were composed of chloroform-methanol and/or *n*-hexane-ethyl acetate in various ratios. On an FT-IR spectrophotometer (Shimadzu, IR Prestige-21, Kyoto, Japan), FT-IR spectra were captured using the CHCl_3_ procedure or neat. Using a tunable multinuclear probe, ^1^H (400 MHz) and ^13^C (100 MHz) NMR spectra were captured in CDCl_3_ solution (Bruker DPX-400 spectrometer, Billerica, USA). With tetramethylsilane (TMS) serving as the internal standard, chemical shifts were reported in ppm units, and *J* values are displayed in Hz.

### 3.2. Synthesis

**Methyl 2,3-di-*O*-stearoyl-α-l-rhamnopyranoside (4) and methyl 3,4-di-*O*-stearoyl-α-l-rhamnopyranoside (5):** To a cooled (0 °C) well-stirred solution of methyl α-l-rhamnopyranoside (**3**) (2.0 g, 11.24 mmol) in anhydrous pyridine (6 mL) was added stearoyl chloride (7.422 g, 24.52 mmol) slowly followed by addition of a catalytic amount of DMAP. It was stirred at this temperature for 4 h and then for 14 h at room temperature when TLC indicated the conversion of the starting compound into two faster-moving products (*R*_f_ = 0.71 and 0.33, chloroform/methanol = 5/1, *v*/*v*). The reaction was stopped by adding a few pieces of ice to the reaction flask and extracting the product with dichloromethane (DCM, 3 × 10 mL). The combined organic (DCM) layer was washed successively with dilute hydrochloric acid (5%), saturated aqueous sodium hydrogen carbonate solution, and distilled water. The organic layer was dried over MgSO_4_, filtered and the filtrate was concentrated under reduced pressure to leave a syrupy mass, which was purified by silica gel column chromatography. Initial elution with chloroform-methanol (20:1, *v*/*v*) provided a higher *R*_f_ (0.68) compound **4** as a syrup (2.249 g, 26%) which resisted crystallization. *R*_f_ = 0.68 (chloroform/methanol = 5/1); FT-IR (neat): 3300–3500 (br, OH), 1744, 1739 (CO), 1065 cm^−1^ (pyranose ring); ^1^H NMR (400 MHz, CDCl_3_): δ_H_ 5.12–5.15 (dd, *J* = 9.6 and 3.2 Hz, 1H, H-3), 5.23 (s, 1H, H-2), 4.61 (s, 1H, H-1), 3.71–3.78 (m, 1H, H-5), 3.60 (t, *J* = 9.6 Hz, 1H, H-4), 3.39 (s, 3H, OC*H*_3_), 2.38 and 2.31 [2 × t, *J* = 7.6 Hz, 4H, 2 × CH_3_(CH_2_)_15_C*H*_2_CO], 1.59–1.66 [m, 4H, 2 × CH_3_(CH_2_)_14_C*H*_2_CH_2_CO], 1.37 (d, *J* = 6.0 Hz, 3H, 6-C*H*_3_) 1.20–1.37 [br m, 56H, 2 × CH_3_(C*H_2_*)_14_(CH_2_)_2_CO], 0.89 [t, *J* = 6.4 Hz, 6H, 2 × C*H*_3_(CH_2_)_16_CO]; ^13^C NMR (100 MHz, CDCl_3_): δ_C_ 172.82, 174.08, (2 × C_17_H_35_*C*O), 98.5 (C-1), 72.1 (C-2), 71.5 (C-3), 69.8 (C-4), 68.4 (C-5), 54.9 (O*C*H_3_), 34.23, 34.24 [2 × CH_3_(CH_2_)_15_*C*H_2_CO], 31.9 (2) [2 × CH_3_(CH_2_)_14_*C*H_2_CH_2_CO], 29.7(7), 29.69(4), 29.68(2), 29.67(2), 29.65, 29.54, 29.48, 29.37(2), 29.32, 29.29, 29.09(2) [2 × CH_3_(CH_2_)_2_(*C*H_2_)_12_(CH_2_)_2_CO], 25.0, 24.7 [2 × CH_3_CH_2_*C*H_2_(CH_2_)_14_CO], 22.7(2) [2 × CH_3_*C*H_2_(CH_2_)_15_CO], 17.5 [6-*C*H_3_], 14.1(2) [2 × *C*H_3_(CH_2_)_16_CO]. The structure and position of the signals were also confirmed by its DEPT-135, 2D COSY, and 2D HMBC experiments.

Further elution with chloroform-methanol (15:1) slowly furnished the lower *R*_f_ compound **5** (3.239 g, 39%) as a semi-solid. *R*_f_ = 0.33 (chloroform/methanol = 5/1); FT-IR (neat): 3320–3480 (br, OH), 1745, 1737 (CO), 1069 cm^−1^ (pyranose ring); ^1^H NMR (400 MHz, CDCl_3_): δ_H_ 5.23–5.26 (dd, *J* = 9.6 and 3.2 Hz, 1H, H-3), 5.13 (t, *J* = 10.0 Hz, H-4), 4.71 (s, 1H, H-1), 4.04 (s, 1H, H-2), 3.83–3.90 (m, 1H, H-5), 3.41 (s, 3H, OC*H*_3_), 2.23–2.36 [m, 4H, 2 × CH_3_(CH_2_)_15_C*H*_2_CO], 1.58–1.70 [m, 4H, 2 × CH_3_(CH_2_)_14_C*H*_2_CH_2_CO], 1.25–1.38 [br m, 56H, 2 × CH_3_(C*H_2_*)_14_(CH_2_)_2_CO], 1.23–1.24 (d, *J* = 6.4 Hz, 3H, 6-C*H*_3_) 0.90 [t, *J* = 6.4 Hz, 6H, 2 × C*H*_3_(CH_2_)_16_CO]; ^13^C NMR (100 MHz, CDCl_3_): δ_C_ 174.2, 172.4, (2 × C_17_H_35_*C*O), 100.3 (C-1), 71.3 (C-3), 70.6 (C-4), 69.6 (C-2), 66.18 (C-5), 55.08 (O*C*H_3_), 34.29(2) [2 × CH_3_(CH_2_)_15_*C*H_2_CO], 31.9 (2) [2 × CH_3_(CH_2_)_14_*C*H_2_CH_2_CO], 29.7(6), 29.68(3), 29.64(3), 29.49(3), 29.38(2), 29.29(3), 29.19(2), 29.16(2) [2 × CH_3_(CH_2_)_2_(*C*H_2_)_12_(CH_2_)_2_CO], 24.9(2) [2 × CH_3_CH_2_*C*H_2_(CH_2_)_14_CO], 22.7(2) [2 × CH_3_*C*H_2_(CH_2_)_15_CO], 17.4 [6-*C*H_3_], 14.1(2) [2 × *C*H_3_(CH_2_)_16_CO]. The structure and position of the signals were also confirmed by its 2D HMBC experiments.

**General procedure for 4-*O*- and 2-*O*-acylation of 4 and 5.** After adding a catalytic amount of DMAP, a cooled (0 °C) solution of the **4** or **5** (0.1 g) in dry pyridine (1 mL) and the appropriate acyl halide (1.1 eq.) was slowly added. The reaction mixture was allowed to reach room temperature while being stirred for another 11 to 15 h. To break down any additional acyl halide, some cold was added to the reaction mixture before it was extracted with DCM (5 × 3 mL). Brine, saturated aqueous sodium hydrogen carbonate solution, and 5% hydrochloric acid were used to wash the DCM layer in that order. Under low pressure, the organic layer was compressed and dried. The resulting thick residue was purified using column chromatography (CC). CC was eluted with a gradient of pure *n*-hexane to *n*-hexane/ethyl acetate = 8/1 and the products were concentrated to afford the corresponding 4-*O*- and 2-*O*-acyllrhamnopyranosides.

**Methyl 4-*O*-pentanoyl-2,3-di-*O*-stearoyl-α-l-rhamnopyranoside (6):** Syrup; Yield 82%; *R*_f_ = 0.54 (*n*-hexane/EA = 5/1, *v*/*v*); FT-IR (neat): 1748, 1747, 1726 (CO), 1081 cm^−1^ (pyranose ring); ^1^H NMR (400 MHz, CDCl_3_): δ_H_ 5.29–5.33 (dd, *J* = 10.0 and 3.2 Hz, 1H, H-3), 5.26 (s, 1H, H-2), 5.10 (t, *J* = 10.0 Hz, 1H, H-4), 4.63 (s, 1H, H-1), 3.85–3.89 (m, 1H, H-5), 3.40 (s, 3H, OC*H*_3_), 2.19–2.45 [3 × m, 6H, 2 × CH_3_(CH_2_)_15_C*H*_2_CO and CH_3_(CH_2_)_2_C*H*_2_CO], 1.53–1.67 [m, 10H, 2 × CH_3_(CH_2_)_13_(C*H*_2_)_2_CH_2_CO) and CH_3_CH_2_C*H*_2_CH_2_CO], 1.25–1.38 [br m, 57H, 2 × CH_3_(C*H*_2_)_13_(CH_2_)_3_CO, CH_3_C*H*_2_(CH_2_)_2_CO, and 6-C*H*_3_)], 0.88–0.91 [t, 9H, 2 × C*H*_3_(CH_2_)_16_CO and C*H*_3_(CH_2_)_3_CO]; ^13^C NMR (100 MHz, CDCl_3_): δ_C_ 172.60, 172.72, 172.92 (2 × C_17_H_35_*C*O, and C_4_H_9_*C*O), 98.6 (C-1), 70.8 (C-2), 69.6 (C-3), 68.9 (C-4), 66.3 (C-5), 55.1 (O*C*H_3_), 34.2, 34.1(2) [CH_3_(CH_2_)_2_*C*H_2_CO and 2 × CH_3_(CH_2_)_15_*C*H_2_CO], 31.9 (4) [2 × CH_3_(CH_2_)_14_*C*H_2_CH_2_CO and CH_3_(*C*H_2_)_2_CH_2_CO], 29.73(6), 29.72(4), 29.71(2), 29.67(1), 29.64(2), 29.53, 29.49, 29.37, 29.33, 29.32, 29.29, 29.17, 29.15, 29.09 [2 × CH_3_(CH_2_)_2_(*C*H_2_)_12_(CH_2_)_2_CO], 25.0, 24.7 [2 × CH_3_CH_2_*C*H_2_(CH_2_)_14_CO], 22.69(2) [2 × CH_3_*C*H_2_(CH_2_)_15_CO], 17.4 [6-*C*H_3_], 14.1(3) [2 × *C*H_3_(CH_2_)_16_CO and *C*H_3_(CH_2_)_3_CO].

**Methyl 4-*O*-decanoyl-2,3-di-*O*-stearoyl-α-l-rhamnopyranoside (7):** Homogeneous syrup; Yield 74%; *R*_f_ = 0.61 (*n*-hexane/EA = 5/1); FT-IR (CHCl_3_): 1745(2), 1736 (CO), 1047 cm^−1^ (pyranose ring); ^1^H NMR (400 MHz, CDCl_3_): δ_H_ 5.29–5.33 (dd, *J* = 10.0 and 3.2 Hz, 1H, H-3), 5.26 (s, 1H, H-2), 5.1 (t, *J* = 9.6 Hz, 1H, H-4), 4.63 (s, 1H, H-1), 3.84–3.91 (m, 1H, H-5), 3.40 (s, 3H, OC*H*_3_), 2.39–2.46 [m, 6H, 2 × CH_3_(CH_2_)_15_C*H*_2_CO and CH_3_(CH_2_)_7_C*H*_2_CO], 2.19–2.30 [2 × m, 4H, CH_3_(CH_2_)_5_(C*H*_2_)_2_CH_2_CO], 1.55–1.65 [m, 12H, 2 × CH_3_(CH_2_)_13_(C*H*_2_)_2_CH_2_CO and CH_3_(CH_2_)_3_(C*H*_2_)_2_(CH_2_)_3_CO], 1.23–1.32 [br m, 61H, 2 × CH_3_(C*H*_2_)_13_(CH_2_)_3_CO, CH_3_(C*H*_2_)_3_(CH_2_)_5_CO, and 6-C*H*_3_)], 0.88–0.91 [t, 9H, 2 × C*H*_3_(CH_2_)_16_CO and C*H*_3_(CH_2_)_8_CO]; ^13^C NMR (100 MHz, CDCl_3_): δ_C_ 172.61, 172.73, 172.93 (2 × C_17_H_35_*C*O and C_9_H_19_*C*O), 98.6 (C-1), 70.8 (C-2), 69.6 (C-3), 68.9 (C-4), 66.3 (C-5), 55.1 (O*C*H_3_), 34.2, 34.1(2) [CH_3_(CH_2_)_7_*C*H_2_CO and 2 × CH_3_(CH_2_)_15_*C*H_2_CO], 31.9(2), 31.8, 31.7 [CH_3_(CH_2_)_5_(*C*H_2_)_2_CH_2_CO and 2 × CH_3_(CH_2_)_14_*C*H_2_CH_2_CO], 29.72(5), 29.71(3), 29.67(2), 29.64(2), 29.5, 29.49(2), 29.37, 29.31, 29.29, 29.19, 29.17(2), 29.15, 29.09, 29.07(2), 28.92 [2 × CH_3_(CH_2_)_2_(*C*H_2_)_12_(CH_2_)_2_CO and CH_3_(CH_2_)_3_(*C*H_2_)_2_(CH_2_)_3_CO], 25.0, 24.9, 24.7(2) [2 × CH_3_CH_2_*C*H_2_(CH_2_)_14_CO and CH_3_CH_2_(*C*H_2_)_2_(CH_2_)_5_CO], 22.6(3) [2 × CH_3_*C*H_2_(CH_2_)_15_CO and CH_3_*C*H_2_(CH_2_)_7_CO], 17.4 [6-*C*H_3_], 14.1(2), 14.0 [2 × *C*H_3_(CH_2_)_16_CO and *C*H_3_(CH_2_)_8_CO].

**Methyl 4-*O*-benzoyl-2,3-di-*O*-stearoyl-α-l-rhamnopyranoside (8):** Pasty-mass; Yield 76%; *R*_f_ = 0.57 (*n*-hexane/EA = 5/1); FT-IR (CHCl_3_): 1748, 1743, 1738 (CO), 1068 cm^−1^ (pyranose ring); ^1^H NMR (400 MHz, CDCl_3_): δ_H_ 7.86–8.06 (m, 2H, Ar-*H*), 7.48–7.59 (m, 2H, Ar-*H*), 7.32–7.46 (m, 1H, Ar-*H*), 5.15–5.49 (dd, *J* = 10.0 and 3.6 Hz, 1H, H-3), 5.41 (s, 1H, H-2), 5.3 (t, *J* = 6.4 Hz, 1H, H-4), 4.63 (s, 1H, H-1), 3.99–4.1 (m, 1H, H-5), 3.45 (s, 3H, OC*H*_3_), 2.1–2.3 [2 × m, 4H, 2 × CH_3_(CH_2_)_15_C*H*_2_CO], 1.57–1.63 [m, 4H, 2 × CH_3_(CH_2_)_14_C*H*_2_CH_2_CO], 1.20–1.39 [br m, 59H, 2 × CH_3_(C*H_2_*)_14_(CH_2_)_2_CO and 6-C*H*_3_], 0.83–0.92 [t, *J* = 6.0 Hz, 6H, 2 × C*H*_3_(CH_2_)_16_CO]; ^13^C NMR (100 MHz, CDCl_3_): δ_C_ 172.9, 172.6 (2 × C_17_H_35_*C*O), 165.6 (C_6_H_5_CO), 133.3, 129.9–129.7(2), 128.5–128.4(2), 128.3 (Ar-*C*), 98.7 (C-1), 71.7 (C-3), 69.8 (C-2), 68.8(C-4), 66.5 (C-5), 55.1 (O*C*H_3_), 34.2 [2 × CH_3_(CH_2_)_15_*C*H_2_CO], 31.9 [2 × CH_3_(CH_2_)_14_*C*H_2_CH_2_CO], 29.7(6), 29.67(4), 29.60(2), 29.56, 29.53, 29.48, 29.36(2), 29.33, 29.30, 29.14, 29.10, 29.07, 28.98, 28.93 [2 × CH_3_(CH_2_)_2_(*C*H_2_)_12_(CH_2_)_2_CO], 24.7, 24.9 [2 × CH_3_CH_2_*C*H_2_(CH_2_)_14_CO], 22.69 [2 × CH_3_*C*H_2_(CH_2_)_15_CO], 17.5[6-*C*H_3_], 14.1 [2 × *C*H_3_(CH_2_)_16_CO].

**Methyl 2-*O*-hexanoyl-3,4-di-*O*-stearoyl-α-l-rhamnopyranoside (9):** Clear-syrup; Yield 81%; *R*_f_ = 0.58 (*n*-hexane/EA = 5/1); FT-IR (neat): 1752, 1749, 1729 (CO), 1051 cm^−1^ (pyranose ring); ^1^H NMR (400 MHz, CDCl_3_): δ_H_ 5.30–5.33 (dd, *J* = 10.0 and 3.6 Hz, 1H, H-3), 5.26 (s, 1H, H-2), 5.1 (t, *J* = 10.0 Hz, 1H, H-4), 4.63 (s, 1H, H-1), 3.84–3.91 (m, 1H, H-5), 3.40 (s, 3H, OC*H*_3_), 2.19–2.42 [3 × m, 6H, 2 × CH_3_(CH_2_)_15_C*H*_2_CO and CH_3_(CH_2_)_3_C*H*_2_CO], 1.55–1.65 [m, 12H, 2 × CH_3_(CH_2_)_13_(C*H*_2_)_2_CH_2_CO) and CH_3_CH_2_CH_2_C*H*_2_CH_2_CO], 1.23–1.32 [br m, 59H, 2 × CH_3_(C*H*_2_)_13_(CH_2_)_3_CO, CH_3_(C*H*_2_)_2_(CH_2_)_2_CO, and 6-C*H*_3_)], 0.88–0.91 [t, 9H, 2 × C*H*_3_(CH_2_)_16_CO and C*H*_3_(CH_2_)_4_CO]. ^13^C NMR (100 MHz, CDCl_3_): δ_C_ 172.93, 172.72, 172.61 (2 × C_17_H_35_*C*O and C_5_H_11_*C*O), 98.6 (C-1), 70.8 (C-2), 69.6 (C-3), 68.9 (C-4), 66.3 (C-5), 55.1 (O*C*H_3_), 34.2, 34.1(2) [CH_3_(CH_2_)_3_*C*H_2_CO and 2 × CH_3_(CH_2_)_15_*C*H_2_CO], 31.9, 31.2 (2) [2 × CH_3_(CH_2_)_14_*C*H_2_CH_2_CO and CH_3_CH_2_CH_2_*C*H_2_CH_2_CO], 29.72(7), 29.67(4), 29.53(3), 29.5(2), 29.37(1), 29.32(3), 29.1(2), 29.09(2) [2 × CH_3_(CH_2_)_2_(*C*H_2_)_12_(CH_2_)_2_CO], 25.0 24.7, 24.6 [2 × CH_3_CH_2_*C*H_2_(CH_2_)_14_CO and CH_3_CH_2_*C*H_2_CH_2_CH_2_CO], 22.69(2), 22.2 [2 × CH_3_*C*H_2_(CH_2_)_15_CO and CH_3_*C*H_2_CH_2_CH_2_CH_2_CO], 17.4 [6-*C*H_3_], 14.1, 13.9(2) [2 × *C*H_3_(CH_2_)_16_CO and *C*H_3_(CH_2_)_4_CO].

**Methyl 2-*O*-decanoyl-3,4-di-*O*-stearoyl-α-l-rhamnopyranoside (10):** Syrup; Yield 89%; *R*_f_ = 0.59 (*n*-hexane/EA = 5/1); FT-IR (CHCl_3_): 1750, 1748, 1741 (CO), 1081 cm^−1^ (pyranose ring); ^1^H NMR (400 MHz, CDCl_3_): δ_H_ 5.29–5.32 (dd, *J* = 10.0 and 3.6 Hz, 1H, H-3), 5.26 (s, 1H, H-2), 5.1 (t, *J* = 9.6 Hz, 1H, H-4), 4.63 (s, 1H, H-1), 3.85–3.89 (m, 1H, H-5), 3.40 (s, 3H, OC*H*_3_), 2.18–2.42 [m, 6H, 2 × CH_3_(CH_2_)_15_C*H*_2_CO and CH_3_(CH_2_)_7_C*H*_2_CO], 1.54–1.67 [m, 10H, 2 × CH_3_(CH_2_)_13_(C*H*_2_)_2_CH_2_CO and CH_3_(CH_2_)_6_C*H*_2_CH_2_CO], 1.23–1.39 [br m, 67H, 2 × CH_3_(C*H*_2_)_13_(CH_2_)_3_CO, CH_3_(C*H*_2_)_6_(CH_2_)_2_CO and 6-C*H*_3_)], 0.88–0.91 [t, 9H, 2 × C*H*_3_(CH_2_)_16_CO and C*H*_3_(CH_2_)_8_CO]; ^13^C NMR (100 MHz, CDCl_3_): δ_C_ 172.9, 172.7, 172.6 (2 × C_17_H_35_*C*O, and C_9_H_19_*C*O), 98.5 (C-1), 70.8 (C-2), 69.5 (C-3), 68.9 (C-4), 66.2 (C-5), 55.1 (O*C*H_3_), 34.2, 34.1(2) [CH_3_(CH_2_)_7_*C*H_2_CO and 2 × CH_3_(CH_2_)_15_*C*H_2_CO], 31.9, 31.8(2), [CH_3_(CH_2_)_6_*C*H_2_CH_2_CO and 2 × CH_3_(CH_2_)_14_*C*H_2_CH_2_CO], 29.72(6), 29.67(3), 29.53(2), 29.49, 29.46, 29.43(2), 29.40(3), 29.37(2), 29.32(2), 29.30(2), 29.28, 29.2, 29.1, 29.0 [2 × CH_3_(CH_2_)_2_(*C*H_2_)_12_(CH_2_)_2_CO and CH_3_(CH_2_)_2_(*C*H_2_)_4_(CH_2_)_2_CO], 25.0, 24.9, 24.7 [2 × CH_3_CH_2_*C*H_2_(CH_2_)_14_CO and CH_3_CH_2_*C*H_2_(CH_2_)_6_CO], 22.7, 22.6(2) [2 × CH_3_*C*H_2_(CH_2_)_15_CO and CH_3_*C*H_2_(CH_2_)_7_CO], 17.4 [6-*C*H_3_], 14.1(3) [2 × *C*H_3_(CH_2_)_16_CO and *C*H_3_(CH_2_)_8_CO].

**Methyl 2-*O*-benzoyl-3,4-di-*O*-stearoyl-α-l-rhamnopyranoside (11):** Pasty-mass; Yield 82%; *R*_f_ = 0.63 (*n*-hexane/EA = 5/1); FT-IR (CHCl_3_): 1753, 1748, 1745, (CO), 1066 cm^−1^ (pyranose ring); ^1^H NMR (400 MHz, CDCl_3_): δ_H_ 7.9–8.0 (m, 2H, Ar-*H*), 7.60–7.62 (m, 2H, Ar-*H*), 7.5–7.4 (m, 1H, Ar-*H*), δ_H_ 5.41–5.44 (dd, *J* = 10.0 and 3.2 Hz, 1H, H-3), 5.48 (s, 1H, H-2), 5.26 (t, *J* = 6.4 Hz, 1H, H-4), 4.8 (s, 1H, H-1), 4.05–4.07 (m, 1H, H-5), 3.45 (s, 3H, OC*H*_3_), 2.1–2.3 [2 × m, 4H, 2 × CH_3_(CH_2_)_15_C*H*_2_CO], 1.57–1.63 [m, 4H, 2 × CH_3_(CH_2_)_14_C*H*_2_CH_2_CO], 1.20–1.39 [br m, 59H, 2 × CH_3_(C*H_2_*)_14_(CH_2_)_2_CO and 6-C*H*_3_], 0.86–0.92 [t, *J* = 6.0 Hz, 6H, 2 × C*H*_3_(CH_2_)_16_CO].

### 3.3. PASS Calculation

The basic structure/geometry of rhamnopyranoside **3** was collected from PubChem (Conformer3D_CID_84695). ChemDraw was used to create the structures of the target compounds **4**–**11**. Following that, these were transformed into the appropriate SD (standard data) file format(s). These SD files were separately utilized with the online version of PASS (prediction of activity spectra for drugs) to predict the biological spectrum. Many programs that are online-oriented and free to use are accessible [35]. Here, an online-built PASS (passonline.pharmaexpert.ru/index.php) is assigned with the goal of predicting the biologic-related features of the rhamnopyranoside-derived **3**–**11** [35,36]. This PASS algorithm is frequently used to forecast around 4000 different biological pursuit types with high precision (over 95%). In this resource, the Pa (probability for an active compound, 0.000 to 1.000) and Pi calculations are shown (probability for an inactive compound, 0.000 to 1.000). For all organic molecules, potential compounds are screened as Pa > Pi (Pa + Pi ≠ 1). Several suggestions for potential biological activity are provided based on Pa values. For instance, Pa > 0.7 is taken to suggest a higher likelihood of discovering a similar activity empirically. Additionally, Pa < 0.5 is seen as the reduced likelihood of obtaining the activity under experimental circumstances. These suggestions were developed using data from more than three million bioactive substances. Based on all of these factors, PASS can be viewed as an intrinsic property of any potential therapeutic molecule [36].

### 3.4. Evaluation of In Vitro Antimicrobial Activities

**(a) Screening of antibacterial efficacy**. The “disc diffusion” method was employed for pure compounds **18** through **27** in 2% DMF solutions, one of the known in vitro antibacterial screening techniques. It was maintained to follow the guidelines established by the Clinical and Laboratory Standards Institute (CLSI) [37]. Bacterial organisms were cultured on Mueller-Hinton (agar and broth) medium. The agar plates with test microorganisms were inoculated at 37 °C for 48 h. The filter paper discs (~6 mm in diameter), containing the synthesized compound at the desired concentration, are placed on the agar surface, followed by incubation. The test compound(s) diffused into the agar. The inhibition of germination and growth organisms was then measured as the diameters of the inhibition zone(s). Each experiment was conducted thrice with proper control (only with DMF). Standard antimicrobial tetracycline was also utilized for comparison and validation purposes. Bacterial and fungal pathogen test tube cultures were obtained from the Department of Microbiology, University of Chittagong, Bangladesh.

**(b) Evaluation of antifungal efficacy**. The “food poisoning” method was used for assessing antifungal susceptibility [37,40]. To put it briefly, the culture of fungi was conducted using sabouraud (agar and broth, PDA) medium. After 3–5 days of incubation, the fungus’ linear mycelial growth was quantified. In general, the following formula was used to determine the % susceptibility of the radial mycelial growth of the fungal species.
$$I=\left\{\frac{\left(C-T\right)}{C}\right\}\times 100$$
where *I* = percentage of inhibition, *C* = diameter of the fungal colony in control (DMF), and *T* = diameter of the fungal colony in treatment. To validate and compare antifungal efficacy, the standard antifungal antibiotic fluconazole (100 μg/mL medium) was tested under similar conditions.

### 3.5. ADME/T Analysis

As discussed in Section 3.3, to collect the isomeric SMILES (simplified molecular-input line-entry system) and SD file formats, the structures of every rhamnopyranoside were accurately drawn in the ChemDraw (3D) application. With the use of these formats, ADMET predictions were possible for all compounds **3**–**11**.

The pkCSM model is said to be the best program to determine various pharmacokinetic property classes in in silico screening procedures [41]. The toxicity of a substance is also determined by this graph-based software as a safety model. Absorption, distribution, metabolism, excretion, and toxicity, sometimes known as ADMET, are key factors in the pharmacological study. The flawed ADMET model is a major barrier in the process of developing new drugs. The prediction of ADMET for each molecule makes it possible to avoid the significant time and cost involved with in vivo laboratory research. For ADMET estimates prior to organic synthesis, computational approaches have become increasingly popular in recent years.

Protein targets known to cause toxic effects and unfavorable medication reactions are known as toxicity targets. Toxicity was measured following the ProTox-II program [43]. The program uses a set of pharmacophores based on protein ligands to forecast potential binding to hazardous targets. The structures of compounds were drawn in this program as mol files and submitted online server to predict five important toxic parameters such as hepatotoxicity, carcinogenicity, immunotoxicity, mutagenicity, and cytotoxicity. Finally, the online SwissADME web server [44] was used to predict the drug-likeness of the molecules.

### 3.6. Molecular Docking

A majority of antifungal drugs (azoles, polyenes, etc.) function by preventing lanosterol 14-demethylase from doing its job (a cytochrome P450 [CYP450] enzyme). As a result, the rhamnopyranoside esters **4**–**11** were molecularly docked with the related protein 3LD6.

**Ligand preparation**. ChemDraw is utilized to accurately represent complicated structures. DFT (RB3LYP, 6-31G, *d*, *p*) is used to optimize each of them [48]. Each compound’s optimized structure is then stored in a PDB file format and used as a ligand.

**Protein preparation**. The chosen three-dimensional (3D, resolution 2.80) crystal structure of human lanosterol 14alpha-demethylase (CYP51, PDB ID: 3LD6) was obtained from the RSCB Protein Data Bank (PDB) [49]. The 3LD6 PDB file is opened in Discovery Studio, followed by the removal of the H_2_O and ketoconazole chains, and saving it as a PDB file. Its energy is then minimized in software called Swisspdb.

**Molecular docking**. The 3LD6 protein’s binding affinity for the rhamnopyranosides and their bonding interaction were determined via molecular docking. Initially, both the ligands and proteins are loaded in the PyMOL software (PyMOL V2.3). A macromolecule, a protein (3LD6), is designated. The energy of ligands (compounds) is reduced, and they are then converted to the respective pdbqt file formats. Both the protein and ligand are forwarded to the AutodockVina wizard with the maximum grid box size to cover the protein’s substrate-binding region. For example, the grid center points were set at X = 38.7754 Å, Y = −1.2253 Å, Z = −3.8249 Å, and the dimension at X = 61.8544 Å, Y = 61.1656 Å, Z = 70.6097 Å. Docking result files are saved and viewed in the BIOVIA Discovery Studio Visualizer 2017 for required 2D and 3D interaction investigations.

## 4. Conclusions

For the first time, the present study described the DMAP-catalyzed dimolar stearoylation of methyl α-l-rhamnopyranoside (**3**). The formation and separation of 2,3-di-*O*- **4** and 3,4-di-*O*-stearates **5** (ratio 2:3) indicated the reactivity of the hydroxylated stereogenic centers of rhamnopyranoside **3** as 3-OH > 4-OH > 2-OH. These stearates and their new six derivatives were well characterized by spectral methods and subjected to in vitro antimicrobial tests. This revealed their better efficacy as antifungals and was supported by PASS and molecular docking studies with lanosterol 14-alpha-demethylase (PDB ID: 4LD6). ADMET-like analysis and their class properties clearly indicated their safer toxicity (toxicity classes 5 and 6). Thus, the present synthesized novel rhamnopyranoside stearates may be an alternative to azole-type antifungals.

## Data Availability

The data presented in this study are available in Appendix A
https://www.mdpi.com/article/10.3390/molecules28030986/s1.

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
