# Peer review of "Rhamnopyranoside-Based Fatty Acid Esters as Antimicrobials: Synthesis, Spectral Characterization, PASS, Antimicrobial, and Molecular Docking Studies"

_molecules, 2023, doi:10.3390/molecules28030986_

Round 1

Reviewer 1 Report

I would recommend the article could be published in Molecules after minor corrections. 

The author needs to address the following comments/corrections.

 1.     Table of content for SI needs page numbers.

2.     All spectra in the SI should include the structure of the molecules.

3.     The author should include the peak peaking for all NMRs.

4.     The author needs to draw the structure with methyl group either Me or CH3 throughout the manuscript.

5.     Yield in the text and in the scheme 1 is different.

6.     Structure elucidation by NMRs could be moved to SI.

Author Response

Thank you for your suggestions and comments. We have corrected the revised manuscript according to your suggestions.

  1. Table of content for SI needs page numbers.

- Page number is added in SI Table.

  1. All spectra in the SI should include the structure of the molecules.

- Structures of compounds are added in SI.

  1. The author should include the peak peaking for all NMRs.

- Thank you. We have changed the 1H NMR spectra of 4, 5, 6, and 7 and added spectra with peak picking values.

  1. The author needs to draw the structure with methyl group either Me or CH3 throughout the manuscript.

- Modified accordingly.

  1. Yield in the text and in the scheme 1 is different.

- We have corrected the same.

  1. Structure elucidation by NMRs could be moved to SI.

- Thank you. We strongly believe that Figure 2 [Confirmation of CO group positions: (A) HMBC spectrum of 4, and (B) HMBC spectrum of 5] has importance for the distinction between compounds 4 and 5. Thus, we prefer to keep it in the main text (such an important figure is also shown in many articles of ‘Molecules’).

Reviewer 2 Report

In this manuscript entitled “Rhamnopyranoside-based fatty acid esters as antimicrobials: Synthesis, spectral characterization, PASS, antimicrobial, and molecular docking studies”, novel rhamnopyranoside-based esters were synthesis were synthesized and their antimicrobial activities were evaluated. Most of the target compounds were fully characterized by FT-IR, 1H, and 13C NMR spectral techniques. In vitro antimicrobial assay and PASS analysis data revealed that fully esterified rhamnopyranosides 6-11 with maximum lipophilic character showed better antifungal susceptibility than antibacterial activity. In order to reveal the action sites of these compounds, molecular docking was also carried out. In general, the research is relatively in-depth and can be accepted for publication in this journal. However, there are still some issues to be solved.

1. NMR spectra, 2D COSY spectrum, and 2D HSQC spectrum of compound 11 were missing.

2. The result showed regioselectivity at C-2, C-3, and C-4 positions, indicating that the reactivities of OH groups in methyl α-L-rhamnopyranoside (3) are 3-OH ˃ 4-OH ˃ 2-OH. Whether other reaction conditions have been tried. Is there a change in the yield ratio of compounds 4 and 5?

3. Why did the author choose three acyl chlorides (C5H11COCl/C9H19COCl/C6H5COCl) for 4-O-acylation of di-O-stearate 4 and 2-O-acylation of di-O-stearate 5?  Why not consider acetyl chloride or acyl chloride containing heterocycles?

4. 3D Interaction of (A) 10, (B) 9, and (C) 7 with amino acid residues of 4LD6 in figure 5 needs to be adjusted. The overlapping names of multiple amino acid residues lead to unclear contents.

Author Response

Comments: In this manuscript entitled “Rhamnopyranoside-based fatty acid esters as antimicrobials: Synthesis, spectral characterization, PASS, antimicrobial, and molecular docking studies”, novel rhamnopyranoside-based esters were synthesis were synthesized and their antimicrobial activities were evaluated. Most of the target compounds were fully characterized by FT-IR, 1H, and 13C NMR spectral techniques. In vitro antimicrobial assay and PASS analysis data revealed that fully esterified rhamnopyranosides 6-11 with maximum lipophilic character showed better antifungal susceptibility than antibacterial activity. In order to reveal the action sites of these compounds, molecular docking was also carried out. In general, the research is relatively in-depth and can be accepted for publication in this journal. However, there are still some issues to be solved.

- Thank you very much for your constructive comments and suggestions. We made changes to the revised manuscript accordingly.

  1. NMR spectra, 2D COSY spectrum, and 2D HSQC spectrum of compound 11 were missing.

- Thank you. We have scanned FT-IR and 1H NMR only of this compound 11 (please see the experimental section), and the structure was also confirmed by analogy with other derivatives (9 and 10). Hence, we are unable to add the 2D spectra of this compound.

  1. The result showed regioselectivity at C-2, C-3, and C-4 positions, indicating that the reactivities of OH groups in methyl α-L-rhamnopyranoside (3) are 3-OH ˃ 4-OH ˃ 2-OH. Whether other reaction conditions have been tried. Is there a change in the yield ratio of compounds 4 and 5?

- Thank you. We have used other reaction conditions to improve the yield only. Hence, we did not scan their spectra.

  1. Why did the author choose three acyl chlorides (C5H11COCl/C9H19COCl/C6H5COCl) for 4-O-acylation of di-O-stearate 4 and 2-O-acylation of di-O-stearate 5?  Why not consider acetyl chloride or acyl chloride containing heterocycles?

- Thank you. We have considered medium- and bulky-acylating agents (available in our laboratory also).

  1. 3D Interaction of (A) 10, (B) 9, and (C) 7 with amino acid residues of 4LD6 in figure 5 needs to be adjusted. The overlapping names of multiple amino acid residues lead to unclear contents.

- Thank you. We have revised it and tried to adjust it. Due to the presence of multiple non-bond interactions, the three-letter symbols of amino-acid residues are merged in Discovery Studio. However, we have improved it.